# Long-Term Survival Rates and Treatment Trends of Burkitt Lymphoma in Patients with HIV—A National Cancer Database (NCDB) Study

**DOI:** 10.3390/cancers16071397

**Published:** 2024-04-02

**Authors:** Clare M. Wieland, Ashley M. Tuin, Elizabeth J. Dort, Alexander G. Hall, Mridula Krishnan, Manasa Velagapudi

**Affiliations:** 1School of Medicine, Creighton University, Omaha, NE 68178, USA; clarewieland@creighton.edu (C.M.W.); ashleytuin@creighton.edu (A.M.T.); elizabethdort@creighton.edu (E.J.D.); 2Department of Clinical Research & Public Health, School of Medicine, Creighton University, Omaha, NE 68178, USA; alexanderhall@creighton.edu; 3Division of Hematology/Oncology, Department of Internal Medicine, University of Nebraska Medical Center, Omaha, NE 68198, USA; mridula.krishnan@unmc.edu; 4Division of Infectious Diseases, Department of Internal Medicine, School of Medicine, Creighton University, CHI Health, Omaha, NE 68124, USA

**Keywords:** Burkitt lymphoma, HIV, NCDB, HIV-associated Burkitt lymphoma, immunotherapy, survival, mortality rate

## Abstract

**Simple Summary:**

AIDS remains a worldwide epidemic, and approximately 1.2 million people are living with HIV in the United States. Burkitt lymphoma accounts for 10–35% of AIDS-defining lymphoma in people with HIV. The aim of our retrospective database study was to assess the long-term survival rates of HIV-associated Burkitt lymphoma on a national level. We found that there was an increased mortality rate of HIV-associated Burkitt lymphoma, and there was a 55% increased risk of death from months 3 to 60 from the time of diagnosis for HIV-associated Burkitt lymphoma compared to patients with Burkitt lymphoma without HIV. Additionally, risk of death was significantly decreased in patients with HIV treated with combination chemotherapy and immunotherapy compared to patients without HIV treated with combination chemotherapy and immunotherapy in the first 3 months following BL diagnosis.

**Abstract:**

Background: Burkitt lymphoma (BL) accounts for 10–35% of AIDS-defining lymphoma in people with HIV (PWH). Previous research consisting of smaller cohorts has shown decreased survival for HIV-associated BL. This study aims to compare overall mortality in BL patients with and without HIV, while investigating impact of treatment modalities in HIV-associated BL. Methods: Using the 2004–2019 NCDB, we identified 4312 patients with stage 3 or 4 BL who had a known HIV status and received either chemotherapy alone or chemotherapy and immunotherapy. Time to death was evaluated using Kaplan–Meier survival estimates. Risk of death was evaluated using an extended multivariable Cox model adjusted for multiple factors and with a Heaviside function for HIV status by time period (0–3 month vs. 3–60 month). Results: Of the 4312 patients included, 1514 (35%) had HIV. For months 0–3 from time of diagnosis, HIV status was not associated with a statistically significant increase in risk of death (HR = 1.04, 95% CI: 0.86, 1.26, *p* = 0.6648). From month 3to 60, positive HIV status was associated with a 55% increase in risk of death compared to those without HIV (95% CI: 1.38, 1.75, *p* < 0.0001). Further, this difference in hazard rates (0–3 vs. 3–60) was statistically significant (HR = 1.49, 95% CI: 1.22–1.82, *p* < 0.001). Conclusions: There is an increased mortality rate from months 3 to 60 in BL patients with HIV compared to patients without HIV. Additionally, risk of death in the first 3 months is significantly decreased by 45% in patients with HIV treated with combination chemotherapy and immunotherapy compared to patients without HIV receiving combination chemotherapy and immunotherapy, providing valuable clinical insight into treatment decision making in the care of HIV-associated BL.

## 1. Introduction

As of 2022, there were 38.4 million people with HIV (PWH) worldwide [1]. In the United States, approximately 1.2 million people are living with HIV [2], and 36,136 individuals in the United States were newly diagnosed with HIV in 2021 [3]. While antiretroviral therapy (ART) in the U.S. is accessible to everyone diagnosed with HIV regardless of CD4 count [4], HIV continues to be a public health crisis [5], and early diagnosis and treatment for HIV and its associated illnesses is imperative.

As the number of people receiving an HIV diagnosis remains high, it is important to remain up to date on the risks and outcomes that this population endures. In addition to opportunistic infections, people with HIV are at increased risk for multiple types of cancer due to the immune dysregulation caused by the illness. Lymphomas account for more than 50% of AIDS-defining cancers [6]. Hodgkin’s lymphoma (HL) and non-Hodgkin’s lymphoma (NHL) are the main types of lymphoma in PWH [7]. While HL in PWH is 5–10 times higher than in the general population and has been increasing [8], NHL is even more common than HL. The most prevalent types of NHL associated with AIDS are Burkitt lymphoma (BL), diffuse large B-cell lymphoma (DLBCL), primary central nervous system lymphoma, plasmablastic lymphoma, and primary effusion lymphoma [9]. Among these subtypes, DLBCL and BL, along with other variants, collectively represent 50%, 40%, and 10% of cases, respectively [10]. The prevalence of non-Hodgkin’s lymphoma in this population goes beyond what could be attributed to non-HIV specific exposures [11]. Additionally, it has been shown that NHL in people with AIDS is more likely to be associated with the Epstein–Barr virus and originates from transformed B-cells [12]. Among those non-Hodgkin’s lymphomas associated with or caused by viral co-infections, Burkitt lymphoma (BL) remains vastly understudied in people with HIV [13].

Burkitt lymphoma accounts for 10–35% of AIDS-defining lymphoma in people with HIV, making it the second most common subtype of non-Hodgkin’s lymphoma occurring in HIV-positive patients with a relatively high CD4 cell count [14]. Burkitt lymphoma (BL) in PWH is 261 times greater than that observed in the general population [15]. The heightened risk of Burkitt lymphoma in this population is explained by multiple mechanisms including immune dysregulation, EBV co-infection, direct effect of the HIV Tat antigen, and multiple other structural proteins on the proliferation of polyclonal B-cells [16]. In addition to being at a higher risk of Burkitt lymphoma, previous studies have shown people with HIV are also more likely to be diagnosed at a more advanced stage, even after accounting for healthcare-related factors [13,17]. This increases the risk of morbidity and mortality and could point to a time-based disease progression that is increased for this population.

Despite the age of antiretroviral therapy (ART), which has greatly extended the life expectancy for people with HIV, HIV continue to be associated with an increased risk of death among patients with lymphoma [18,19]. Previous research has shown decreased survival for HIV-associated Burkitt lymphoma; however, these regional studies consisted of relatively small cohorts of patients (*n* < 300) [20,21,22].

The treatment approach for HIV-associated Burkitt lymphoma mirrors that of individuals without HIV and has also greatly improved over the past decades. Chemotherapy remains the cornerstone of treatment for AIDS-associated Burkitt lymphoma (BL), significantly improving survival outcomes. However, the advent of immunotherapy, particularly with the introduction of anti-CD20 monoclonal antibodies such as rituximab, has demonstrated promising results in enhancing patient survival rates [23,24]. A study of the AIDS Malignancy Consortium 034 trial in 2018, which included patients with diffuse large B-cell lymphoma, indicated that chemoimmunotherapy alongside concurrent antiretroviral therapy (ART) is not detrimental and facilitates expedited immune restoration [25].

To date, no study has isolated the impact of Burkitt lymphoma in people with HIV on a national level. A broader analysis at this level would allow us to better understand and visualize trends in treatment and risk in this population. This study aims to provide an updated and more expansive view of the overall mortality risk and effect of treatment modalities on survival and predictors of survival in HIV-associated Burkitt lymphoma using the National Cancer Database (NCDB).

## 2. Methods

### 2.1. Data Source

Data for this project were abstracted using the 2020 nodal and extranodal lymphoma participant user files (PUFs) from the National Cancer Database (NCDB) which contain survival data for patients diagnosed from 2004 through 2019. NCDB data are deidentified hospital-based HIPAA-compliant data sets, comprising cases submitted to the Commission on Cancer (CoC) of the American College of Surgeons and the American Cancer Society. Within the United States, NCDB PUFs capture approximately 72% of newly diagnosed cases of cancer from more than 1500 commission-accredited cancer programs [26]. This study was reviewed and acknowledged as not human subjects research by the Creighton University IRB (InfoEd ref # 2004080).

### 2.2. Study Cohort

Patients with Burkitt lymphoma were identified using the *International Classification of Diseases for Oncology*, third edition, histology code 9687. HIV status was identified using lymphoma site-specific factor 1 for patients diagnosed before 2018 [27] and using North American Association of Central Cancer Registries item SSDI #3859 for patients diagnosed in 2018 or later [28]. Given the absence of cancer-specific mortality in the NCDB, we only considered patients with advanced cancer stages 3 and 4 to increase the likelihood the mortality outcome represents mortality due to cancer (Figure 1). Additionally, we excluded patients who did not receive either chemotherapy alone or chemotherapy and immunotherapy, patients with an unknown HIV status, and patients missing age, sex, race, or ethnicity.

### 2.3. Statistical Analysis

Patient descriptive statistics were stratified by the presence of comorbid HIV. Continuous variables are presented as median and IQR and compared using the Mann–Whitney test, while categorical variables are presented as frequency and percent and compared using chi-squared tests. The primary aim was to assess long-term survival rates of Burkitt lymphoma in patients living with or without HIV. In order to compare overall survival between patients with or without HIV, time to death and risk of death were evaluated. Time to death was evaluated using Kaplan–Meier survival estimates with time-to-death differences between HIV and non-HIV patients assessed via the log-rank test. Risk of death was evaluated with Cox proportional hazards; specifically, marginal models were used to account for the correlation inherent to patients treated within the same facility. The multivariable Cox proportional hazard model was adjusted for several factors, including extranodal vs. nodal, treatment (chemotherapy and immunotherapy vs. immunotherapy alone), age, sex, white vs. other, and Hispanic vs. not Hispanic. The proportional hazards assumption was assessed graphically using log–negative-log survival plots and found to be untenable with respect to HIV status. As such, a Heaviside function and an extended Cox model for HIV status by time period were estimated; negative log-likelihood and AIC were compared across different initial period cut points to identify the most appropriate fit for the Heaviside function, which was determined to be at month 3—as such, 0–3 and >3–60-month periods were used. Survival was right-censored at 60 months because HIV treatment shifted from 2015 on, signifying a maximum follow-up using this modality of five years (60 months). All analyses were conducted using Statistical Analysis Software (SAS) v. 9.4, and statistical significance was determined using two-tailed *p* < 0.05.

## 3. Results

Between 2004 and 2019, there were a total of 4312 patients meeting the inclusion criteria, with 1514 (35%) having HIV. The majority of these patients were white (80%), male (77%), and non-Hispanic (87%). Most had private insurance (52%) and received chemotherapy alone (66%). When compared to patients who were HIV-negative, patients with HIV were significantly younger, were less likely to be female, and were more likely to be non-white, Hispanic, or uninsured and were less likely to be treated with immunotherapy (Table 1). Additionally, descriptive mortality rates during the study period are provided in Table 2.

### Mortality and Survival

Patients were followed for a median of 33 months (IQR: 7–60 months), with 1875 (43.5%) suffering all-cause mortality and 2437 (65.5%) being censored. Of the censored patients, 1569 (64.4%) had at least 60 months of follow-up. Five-year survival estimates are provided in Table 2, and Figure 2 shows the Kaplan–Meier survival curves stratified by HIV status. Table 3 displays the adjusted hazard ratio for covariates included in the Cox model. The HIV status variable is presented for 0–3 months and >3–60 months due to the violation of the proportional odds assumption; this difference in hazard between time periods (0–3 vs. >3–60 months) was statistically significant (*p* < 0.0001). HIV status was not associated with a statistically significant increase in risk of death for the first 3 months (HR = 1.04, 95% CI: 0.86, 1.26, *p* = 0.6648) but was associated with a 55% increase in risk of death from three month to five years (95% CI: 1.38, 1.75, *p* < 0.0001). When evaluating whether the effect of treatment differed by HIV status, there was a statistically significant interaction effect observed in the first 3 months (interaction *p* = 0.005). Specifically, in the first three months, there was no statistically significant difference in risk of death between patients with or without HIV who received chemotherapy (HR: 1.16, 95% CI: 0.95–1.42, *p* = 0.141); however, patients with HIV who received chemotherapy and immunotherapy averaged 45% lower risk of death compared to non-HIV patients who received chemotherapy and immunotherapy (95% CI: 9% lower to 67% lower, *p* = 0.019). The interaction effect was not statistically significant for >3–60 months (interaction *p* = 0.461), indicating that the 55% increased risk of death associated with HIV reported above was applicable whether the patient received chemotherapy with or without immunotherapy.

## 4. Discussion

Our results indicate that HIV status did not significantly increase hazard of death in first 0–3 months, but it did by 55% in the following 3–60 months (*p* < 0.0001). When compared to a 2021 study (*n* = 249) that found a small difference in overall survival and 3-year follow-up survival (66% vs. 61%) [21], this increased risk is considerable. Similarly, another NCDB study using data from 2004 to 2011 found a 46% increase in hazard of death of HIV-associated Burkitt lymphoma [18]. Our findings include this same patient cohort in addition to more recently diagnosed patients and indicate that while the presence of HIV does not change the early outcomes of a Burkitt lymphoma diagnosis, it is associated with increased risk of death as the disease progresses. This is clinically relevant for both early detection and aggressive treatment of both HIV and Burkitt lymphoma [29].

A possible explanation for this Increased all-cause mortality risk in people with HIV during 3–60 months is the higher rate of infection during induction chemotherapy treatment. A 2021 study demonstrated almost 60% of HIV-positive patients with lymphoma experienced infections during chemotherapy. Independent infection risk factors include the following: patients with a higher number of chemotherapy cycles, grade 4 decrease in neutrophil counts, and less than 6-month duration of HAART at time of diagnosis [30]. The risk of infection particularly increases throughout the treatment course for people with HIV due to the combination of myelosuppression and potentially already reduced CD4 counts. This immunosuppressed state could explain, in part, the increased mortality rate among HIV-positive patients as the disease progresses. Immunosuppression has also been known to be directly linked to AIDS-defining malignancies, Kaposi sarcoma, and non-Hodgkin lymphoma and could be linked to accelerated disease progression [11]. This could provide more illumination of the difference between the 0–3-month mortality risk and the 3–60-month mortality risk for people with HIV. Immune dysregulation in people with HIV, even those with higher CD4 counts, could potentiate a weaker immune response to Epstein–Barr virus-positive Burkitt lymphoma that remains asymptomatic for longer, leading to greater mortality risk and a more advanced stage at diagnosis [11].

The overall survival rates for Burkitt lymphoma have significantly improved over time, rising from 10–40% in the pre-antiretroviral therapy (ART) era to 70–80% in the current era, highlighting advancements in HIV treatment [31]. While antiretroviral therapy has played a crucial role in enhancing survival, recent studies from the National Cancer Database reveal that individuals with HIV and lymphoma still face challenges in survival, even with the adoption of ART [32]. In our study, we also analyzed the survival rates during different periods of ART implementation to better understand the impact on survival in this group. Beginning in 2015, universal ART and widespread use of potent integrase inhibitor-based regimens could be administered concurrently with chemotherapy without major interactions [25,33]. Surprisingly, our data showed that despite the widespread use of this treatment modality, this did not result in significant differences in survival for individuals with HIV-associated Burkitt lymphoma. This finding further suggests that while overall survival has improved, there are additional factors, such as infection risk as outlined above, and treatment modalities as mentioned below are the key factors that influence survival outcomes more than HIV-specific features.

Another outcome studied was the impact of the various treatment modalities on overall survival. Patients were categorized as having received chemotherapy, immunotherapy, both, or neither. We found that receiving chemotherapy was associated with an adjusted 41% increase in the risk of death when compared with those who received both immunotherapy and chemotherapy, supporting the conclusions of previously published studies [34,35,36,37,38,39]. Our study confirmed improved outcomes with combination immunotherapy and chemotherapy in the HIV-positive subpopulation of Burkitt lymphoma, which has not been extensively studied before. Of particular interest, we found that the risk of death in HIV-positive patients is 45% lower for those receiving combination chemotherapy and immunotherapy in the first 3 months from time of diagnosis. Given that there was no statistical difference in months 3–60 based on treatment modality, this indicates that there may be some protective aspect to HIV-positive patients receiving chemotherapy and immunotherapy within the first 3 months from diagnosis. Despite this, our results also show that patients with HIV received immunotherapy at a lower rate than patients without HIV (Table 1). A 2012 study found that there were safety concerns with the use of rituximab in PWH with CD4 counts less than 50/mm^3^, which could possibly explain the less frequent usage of immunotherapy in PWH compared to patients without HIV [40]. Further, a study by the AIDS Malignancy Consortium found that immunotherapy may be associated with improved outcomes in HIV-associated NHL when excluding patients with a CD4 count less than 50/mm^3^ who had an increased risk of death due to infectious causes [41]. Other than what is reported in the literature, due to database limitations, we were unable to further investigate what factors may have contributed to PWH receiving immunotherapy at a decreased rate compared to patients without HIV, despite a decreased risk of death within the first 3 months after diagnosis. These findings underscore the importance of integrating immunotherapeutic approaches into the management of AIDS-associated BL to optimize treatment outcomes and enhance patient survival.

Additionally, previous studies have also demonstrated that people living with material deprivation (i.e., those in lower income quartiles) are more likely to participate in risky behavior, leading to higher chances of HIV infection [42]. Further, patients with lower socioeconomic status have been shown to have decreased access to healthcare [42], which may play a role in the increased mortality of HIV-associated Burkitt lymphoma. Economic issues leading to difficulties maintaining employment, homelessness, lack of necessities, and sex exchange often result in nonadherence to treatment and greater subsequent hazard of death [43]. A limitation of this study is that we were not able to include insurance status as a covariate due to small sample sizes between insurance groups. This should be further elucidated in future studies, assuming that sample sizes are sufficient.

The strengths of this study center on its sample size and time period. We were able to look at a large national sample through the use of the NCDB on a relatively rare and specific subset of two very prevalent disease states of Burkitt lymphoma and HIV. This elucidated more specific characteristics of HIV and BL that were not yet ascertained by more generalized studies. Additionally, in looking at a period of 16 years between 2004 and 2020, we were able to gain the most recent data and track changes over a longer period of time to more effectively study survival in relation to advancements in ART and impact of treatment modalities for HIV-associated Burkitt lymphoma on survival.

The limitations of this study were primarily driven by the data available in the NCDB. The NCDB does not include data about CD4 counts, HIV viral load, or individual drug regimens that constitute chemotherapy and immunotherapy regimens for Burkitt lymphoma. The NCDB also did not include treatment duration or intensity, nor any details on remission. The NCDB does provide the Charleson comorbidity index for each patient; however, the presence of AIDS disproportionately affects the overall score compared to the point value of other common comorbidities. This would likely lead to a skewed sample if this score was to be used in analysis; therefore, we chose to omit this variable. Additionally, future research could include treatment trends and survival outcomes of CNS involvement in HIV-associated Burkitt lymphoma, for which not enough data were available by NCDB. We were only able to calculate the overall unadjusted all-cause mortality rate, as opposed to cancer-specific mortality, in our study due to the absence of that data in the database. Further studies of this magnitude relating to treatment-specific and remission-specific outcomes are needed.

## 5. Conclusions

There is an increased mortality rate among Burkitt lymphoma patients with HIV compared to patients without HIV, supporting the previously published literature. The risk of death is significantly increased in months 3–60 from the time of diagnosis. Notably, there was no statistical difference in risk of death within the first 3 months of diagnosis for Burkitt lymphoma patients regardless of HIV status. Additionally, risk of death within the first 3 months of diagnosis is significantly decreased in HIV-positive patients treated with combination chemotherapy and immunotherapy, providing valuable clinical insight into treatment decision making in the care of HIV-associated Burkitt lymphoma. Our findings emphasize the role of combination therapy administered earlier following diagnosis to promote better outcomes in this population.

## Figures and Tables

**Figure 1 cancers-16-01397-f001:**
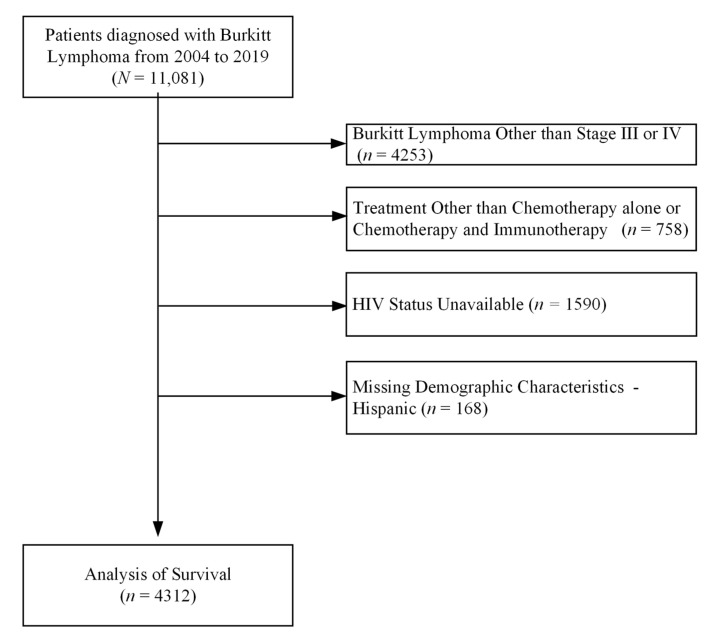
Flowchart of included Burkitt lymphoma (BL) patients and relevant exclusions.

**Figure 2 cancers-16-01397-f002:**
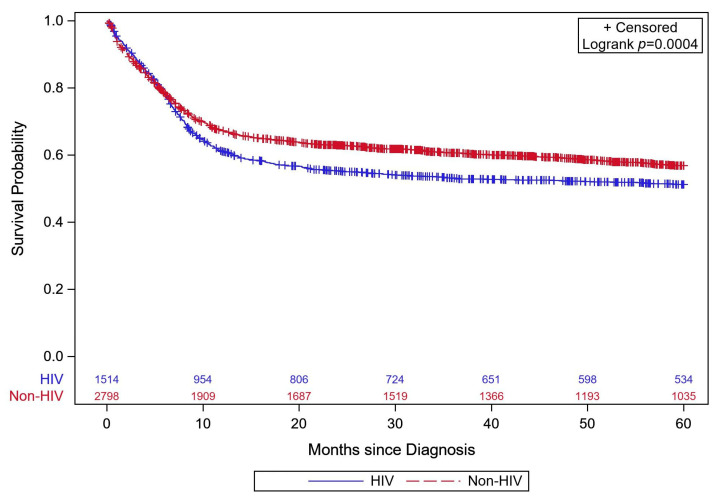
Kaplan–Meier survival curves for patients with stage 3 or 4 Burkitt lymphoma stratified by HIV status from 0 to 60 months post-diagnosis (log-rank *p* < 0.001). Data from the 2004 to 2019 NCDB.

**Table 1 cancers-16-01397-t001:** Demographic characteristics of cohort of patients from 2004 to 2019 with Burkitt lymphoma characterized by HIV status.

*N* = 4312	Overall	HIV Negative	HIV Positive	*p*
Age ^1^	48 (36, 60)	54 (37, 66)	43 (35, 50)	<0.0001
Female	997 (23.1%)	799 (28.6%)	198 (13.1%)	<0.0001
Race				<0.0001
White	3448 (80.0%)	2444 (87.4%)	1004 (66.3%)	
Black	613 (14.2%)	176 (6.3%)	437 (28.9%)	
Other	251 (5.8%)	178 (6.4%)	73 (4.8%)	
Hispanic	567 (13.2%)	292 (10.4%)	275 (18.2%)	<0.001
Primary Payer				<0.001
Not Insured	320 (7.4%)	151 (5.4%)	169 (11.2%)	
Private	2221 (51.5%)	1481 (52.9%)	740 (48.9%)	
Medicaid	734 (17.0%)	317 (11.3%)	417 (27.5%)	
Medicare	902 (20.9%)	758 (27.1%)	144 (9.5%)	
Other Gov.	57 (1.3%)	42 (1.5%)	15 (1.0%)	
Unknown	78 (1.8%)	49 (1.8%)	29 (1.9%)	
Stage of Disease				
3	697 (16.2%)	457 (16.3)	240 (15.9)	0.682
4	3615 (83.8%)	2341 (83.7)	1274 (84.2)	
Extranodal Disease	914 (21.2%)	652 (23.3%)	262 (17.3%)	<0.001
Treatment				<0.001
Chemo Alone	2848 (66.0%)	1745 (62.4%)	1103 (72.9%)	
Chemo and Immuno	1464 (34.0%)	1053 (37.6%)	411 (27.2%)	
Metastatic Brain Involvement ^2^	32 (3.0%)	22 (2.9%)	10 (3.3%)	0.7011

^1^ Age is presented as median and interquartile range and compared statistically using Mann–Whitney test. ^2^ Metastatic brain involvement at time of diagnosis is available from 2016 to 2020; the older variable “CS_METS_DX_BRAIN” available from 2010 to 2015 did not have any recorded cases.

**Table 2 cancers-16-01397-t002:** Five-year survival estimates of patients diagnosed with Burkitt lymphoma from 2004 to 2019 stratified by HIV status.

*N* = 4312	Overall	HIV Negative	HIV Positive	*p*
Overall	54.9 (53.3–56.4)	56.8 (54.9–58.7)	51.2 (48.6–53.8)	<0.001
Age				
<50	62.7 (60.6–64.7)	72.1 (69.4–74.7)	52.9 (49.9–55.8)	<0.001
50+	45.9 (43.6–48.2)	45.8 (43.3–48.3)	46.3 (41.1–51.3)	0.958
Sex				
Male	54.1 (52.3–55.8)	55.6 (53.3–57.9)	51.6 (48.8–54.4)	0.019
Female	57.6 (54.3–60.7)	59.8 (56.2–63.2)	48.5 (41.0–55.5)	0.007
Race				
White	55.1 (53.4–56.8)	56.3 (54.3–58.3)	52.1 (48.9–55.2)	0.027
Black	51.5 (47.3–55.5)	58.6 (50.8–65.7)	48.5 (43.6–53.3)	0.025
Other	60.0 (53.2–66.2)	62.2 (54.1–69.4)	54.7 (41.9–65.8)	0.174
Ethnicity				
Not Hispanic	54.0 (52.3–55.7)	55.9 (53.9–57.9)	50.1 (47.3–53.0)	0.001
Hispanic	60.7 (56.4–64.8)	64.9 (58.8–70.3)	56.3 (50.1–62.1)	0.031
Primary Payer				
Not Insured	51.8 (46.0–57.3)	53.9 (45.3–61.6)	50.2 (42.2–57.6)	0.370
Private	62.0 (59.8–64.0)	65.0 (62.4–67.4)	55.9 (52.1–59.4)	<0.001
Medicaid	54.2 (50.4–57.9)	62.5 (56.6–67.8)	48.1 (43.1–52.9)	<0.001
Medicare	40.1 (36.7–43.4)	40.0 (36.4–43.6)	40.3 (32.0–48.3)	0.697
Other Gov.	53.6 (38.6–66.4)	57.1 (39.0–71.7)	44.4 (18.5–67.7)	0.342
Unknown	45.4 (33.6–56.7)	46.8 (31.8–60.4)	42.8 (23.0–61.2)	0.833
Stage of Disease				
3	67.8 (64.1–71.2)	69.4 (64.8–73.6)	64.6 (58.0–70.5)	0.245
4	52.4 (50.7–54.1)	54.4 (52.2–56.4)	48.7 (45.9–51.5)	0.001
Involvement				
Extranodal	54.6 (52.8–56.3)	56.0 (53.8–58.2)	52.0 (49.1–54.8)	0.025
Nodal	56.2 (52.8–59.4)	59.6 (55.6–63.4)	47.5 (41.2–53.5)	<0.001
Treatment				
Chemo Alone	49.9 (48.0–51.7)	51.8 (49.4–54.2)	46.9 (43.8–49.8)	0.020
Chemo and Immuno	65.1 (62.4–67.6)	65.8 (62.6–68.8)	63.3 (58.1–68.0)	0.379
Metastatic Brain Involvement				
No	54.9 (53.3–56.4)	56.9 (54.9–58.8)	51.1 (48.5–53.8)	<0.001
Yes	58.2 (38.8–73.3)	52.9 (29.8–71.5)	70.0 (32.9–89.2)	0.395

Note. Data presented as percent (95% confidence interval).

**Table 3 cancers-16-01397-t003:** Risk of death—piecewise (HIV status) hazard model with additional time-invariant predictors.

Parameter	Hazard Ratio	95% CI		*p*
HIV vs. NO HIV 0–3 Months	1.04	0.86	1.26	0.6668
HIV vs. NO HIV 3–60 Months	1.55	1.38	1.75	<0.0001
Chemo and Immuno vs. Chemo Only	0.59	0.53	0.65	<0.0001
Extranodal vs. Nodal	0.93	0.83	1.04	0.2083
Age (1 year change)	1.03	1.02	1.03	<0.0001
Female vs. Male	0.86	0.77	0.96	0.0090
Black vs. White	1.11	0.97	1.27	0.1300
Other vs. White	1.00	0.81	1.23	0.9922
Hispanic vs. Not Hispanic	0.84	0.73	0.98	0.0231

Note: the difference in slope coefficient between HIV 0–3 Months and HIV 3–60 is significant, *p* < 0.0003.

## Data Availability

The data sets generated during and/or analyzed during the current study are available in the National Cancer Database repository, https://www.facs.org/quality-programs/cancer-programs/national-cancer-database/ (accessed on 21 July 2023).

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
