# Peer review of "Long-Term Survival Rates and Treatment Trends of Burkitt Lymphoma in Patients with HIV—A National Cancer Database (NCDB) Study"

_cancers, 2024, doi:10.3390/cancers16071397_

Round 1

Reviewer 1 Report

Comments and Suggestions for Authors

Clare Wieland et al. analyse in their article the long term survival rate of HIV infected patients with Burkitt Lymphoma in comparison with infection free patients with the same disease. 

The result is that within the first three months after diagnosis there is no difference in survival rate however in the following 5 years HIV infected patients incur a 55% higher risk to die of the Burkitt Lymphoma. The authors speculate that this might be partially due to the higher infection risk. HIV as such reduces effectiveness with which the immune system can fight infections. This in combination with the use of cytostatic drugs to combat the Lymphoma may lead in the long term to a reduced survival rate. Additionally the Burkitt Lymphoma itself can be associated with an Epstein-Barr virus infection which in HIV patients can lead to a more severe course of the disease. 

The article is based on a large patients group which was well selected (see figure 1). A total of 4300 patients was selected, 35% with a documented HIV infection. These large cohorts make the study unique. 

The article is well structured and its contents well understandable. The presented data support the conclusions (see figure 2). 

I think the article can be published as it is. 

Author Response

Dear Reviewer,
Thank you for your valuable feedback regarding our manuscript, “Long-term survival rates and treatment trends of Burkitt lymphoma in patients with HIV – A National Cancer Database (NCDB) study”

Please see our point-by-point responses below:
There were no changes suggested - thank you for taking the time to review our
manuscript.

Reviewer 2 Report

Comments and Suggestions for Authors

The authors analyzed the survival in HIV-positive and -negative patients with Burkitt lymphoma and also compared different types of treatment.

Specific Points of Criticism and Suggestions for Alterations:

(1)  Line 18:  In the context it is clear. Just to avoid misunderstandings (in the "simple summary"):  "... compared to patients with Burkitt lymphoma without HIV".

(2)  Introduction or Discussion:  Some additional information might be of interest for the reader:  what are the pecentages of HIV+ patients for other lymphomas, such as Hodgkin lymphoma (HL), diffuse large B-cell lymphoma (DLBCL), follicular lymphoma (FL) and others? This information could be presented in a small table or even better in a small figure (with columns each for HL, DLBCL, FL, etc.). Furthermore, the reason for the highest percentage of HIV-positivity in BL is of interest - why? the theory trying to explain these unusual findings might be presented in 1-3 sentences (1-3 references might be sufficient). 

(3)  Figure 2:  Are the survival curves for HIV-neg und HIV-pos statistically significantly different? That could be noted in the legend of this figure.

(4)  Line 190:  "Survival rates at which time point? (5-years, 10-years ...).

(5)  Lines 217-221 and lines 239-248:  There are particuarly many mistakes in these paragraphs. But the whole manuscript should be screened thoroughly for typing errors.

(6)  Any outlook? Any suggestions how to improve outcome?

Comments on the Quality of English Language

English is okay, but there are (too) many typing mistakes, rigorous checking needed.

Author Response

Dear Reviewer,
Thank you for your valuable feedback regarding our manuscript, “Long-term survival rates and treatment trends of Burkitt lymphoma in patients with HIV – A National Cancer Database (NCDB) study”

Please see our point-by-point responses below:
1. Line 18:  In the context it is clear. Just to avoid misunderstandings (in the "simple summary"):  "... compared to patients with Burkitt lymphoma without HIV".
a. This has been updated, thank you.

2. Introduction or Discussion:  Some additional information might be of interest for the reader:  what are the pecentages of HIV+ patients for other lymphomas, such as Hodgkin lymphoma (HL), diffuse large B-cell lymphoma (DLBCL), follicular lymphoma (FL) and others? This information could be presented in a small table or even better in a small figure (with columns each for HL, DLBCL, FL, etc.). Furthermore, the reason for the highest percentage of HIV-positivity in BL is of interest - why? the theory trying to explain these unusual findings might be
presented in 1-3 sentences (1-3 references might be sufficient).

a. Thank you for the suggestion. In paragraph #2& 3 in introduction, we have
included the suggested information with needed references. Please refer to
the following highlighted in the manuscript: “Lymphomas account for more
than 50% of AIDS defining cancers. Hodgkin’s lymphoma (HL) and nonHodgkin’s lymphoma (NHL) are the main types of lymphoma in PWH. HL in
PWH is 5–10 times higher than in the general population and has been
increasing. NHL is more common than HL. The most prevalent types of nonHodgkin lymphoma (NHL) associated with AIDS are Burkitt's lymphoma,
diffuse large B-cell lymphoma, primary central nervous system lymphoma,
plasmablastic lymphoma, and primary effusion lymphoma. Among these
subtypes, DLBCL and BL, along with other variants, represent 50%, 40%, and
10% of cases, respectively...Burkitt lymphoma (BL) in PWH is 261 times
greater than that observed in the general population. The heightened risk of
Burkitt lymphoma in this population is explained by multiple mechanisms
including immune dysregulation, EBV coinfection, direct effect of the HIV Tat
antigen and multiple other structural proteins on the proliferation of
polyclonal B-cells.”

3. Figure 2:  Are the survival curves for HIV-neg und HIV-pos statistically significantly different? That could be noted in the legend of this figure.
a. Thank you for bringing this to our attention. Yes, they differ with p < .001. This
has been noted in the footnote/caption of Figure 2.

4. Line 190:  "Survival rates at which time point? (5-years, 10-years ...).
a. This is an excellent point to clarify. It's changed to “overall survival” as in the
cited article and highlighted in manuscript. This was taken from a review
paper and the citation has been changed to the original publication.

5. Lines 217-221 and lines 239-248: There are particularly many mistakes in these
paragraphs. But the whole manuscript should be screened thoroughly for typing
errors.
a. The typing errors were addressed and the whole manuscript was screened.

6. Any outlook? Any suggestions how to improve outcome?
a. Our recommendations based on this study were added to the conclusion of
the paper (see highlighted). “Our findings emphasize the role of combination
therapy administered earlier following diagnosis to promote better outcomes
in this population.” 

Reviewer 3 Report

Comments and Suggestions for Authors

The study has shown long-term follow-up of BL. The authours need to to include what regimens were used in the treatment of HIV+ BL.

Author Response

Dear Reviewer,
Thank you for your valuable feedback regarding our manuscript, “Long-term survival rates and treatment trends of Burkitt lymphoma in patients with HIV – A National Cancer Database (NCDB) study”

Please see our point-by-point responses below:
1. The study has shown long-term follow-up of BL. The authors need to include what regimens were used in the treatment of HIV+ BL.

a. Unfortunately, this specific data is not available to us through the National
Cancer Database. We have spoken generally about the regimens, involving anti-retroviral therapy, chemotherapy and immunotherapy, but cannot get more specific than that. 

Reviewer 4 Report

Comments and Suggestions for Authors

The authors use the National Cancer Database (US) 2004-2019 to examine mortality among patients with stage 3 or 4 Burkitt lymphoma by HIV status.  HRs reported for 0-3m and 3-60m show a difference in mortality for the later period.

Abstract

Stage 3 or 4 BL treated with chemo either alone or with immunotherapy reported in abstract- were these the only cases to be included?

In Abstract, the aim is stated as looking at BL mortality.  Please be clear in the abstract whether you are considering overall survival (i.e. death from any cause) or BL-specific survival (i.e. death attributable to BL).

Please report the HR for the total 5y period so that comparisons can be made to other studies.

In abstract, when reporting HR for HIV association with death, please state that this is compared to those without HIV. 

Conclusion of the abstract states there was a difference in risk of death between those treated with chemo & immunotherapy compared to those only on chemo.  Is this for all patients or is there a difference between those with HIV and those without- or indeed are HIV patients doing as well as non-HIV patients on the different treatments?  Please report these results in the abstract.  The introduction suggests that immunotherapy may be beneficial in HIV-BL but this aim is not reflected on in the abstract.

Simple summary

Please check that the summary is consistent with the aims of the study (eg treatment trends were not examined).  Please change the time period 3-50m to 3-60m.  Please amend the last sentence in accordance with the comment above – the results are reported on in total and have not examined whether HIV and non-HIV-BL patients have different mortality within treatment modalities.

Introduction

Mention ART in first paragraph-and in the context of US, where treatment accessible

The introduction would be clearer if it focused on BL. (the info at the start of the second paragraph on NHL and HL is not necessary) 

Some more information on treatment of BL and how this has changed- when was rituximab introduced, when was it given to HIV-BL cases?  Is this information only from trials- what is the real-world picture, or is this limited?

Methods

Flow chart- could the flow include the numbers included at each stage of exclusion, so it can be seen how many individuals have stage 3 or 4 BL, how many of those were treated with chemo &/or immunotherapy, etc?

Cox regression model included comparisons of Black to Whites, Other ethnicities to White but compared Hispanics to not Hispanics.  It would be better to compare Hispanics to Whites also.

Extranodal vs nodal disease were included in the Cox model but these data are not described in Table 1.

Results

Table 1- add a total column to describe the whole cohort of treated stage 3 & 4 BLs.  Add the overall survival and 95% confidence intervals for each group; for Age, this could be given for 2 or 3 sensible age bands; for Sex, include for Males as well as Females.  Include numbers for Extra- vs nodal disease.

Also add to the first paragraph that those with HIV were less likely to be treated with immunotherapy.

Paragraph 2 states that the difference in hazard between time periods 0-3m and 3-60m was statistically significant- not sure that the reporting of HR comparing the periods is helpful and would be better to report the test for proportionality between the HIV and non-HIV survival curves.  Please report the cutpoint on the intervals as either, eg: 0-3m and >3-60m; or 0-<3m and 3-60m as appropriate.  What is the justification for choosing 3 months as the cutpoint; while the Kaplan-Meier curves suggest divergence of the survival curves at 7-8 months? 

Figure 2- please label the x-axis as months since diagnosis.  Please modify the title to include that these are survival curves for stage 3 or 4 Burkitt lymphoma, and add the data source (ie NCDB) and the years of diagnosis (2004-2019).  Please also tidy the figure by removing the title “Product-Limit Survival Estimates”, the title “HIV” from the legend and instead use the labels “non-HIV” instead of 0 and “HIV” instead of 1.

There is no comparison of the HIV and non-HIV patients within treatment groups which would be a useful addition to examine whether HIV patients on immunotherapy do as well as the non-HIV patients.  If this is something the study could explore, descriptions of the patients within treatment groups by HIV status should be shown too.

Discussion

The authors mention a previous analysis of HIV-BL mortality using the NCDB for the years 2004-2011.  The present study will cover some of the same patients- more should be said on the overlap, and the HR for patients from 2012-2019 presented.

The authors later state (line 197) that from 2015 onwards the universal ART and integrase inhibitor-based regimens could be administered with chemo but that their data did not show differences in HIV-BL survival with this treatment modality.  Have the authors examined this?  There were no analyses restricted to 2015 onwards, nor of examining HIV-BL survival by treatment.

Paragraph on lines 205-214 reports that chemo only compared to chemo & immunotherapy increase the risk of death by 41% and then it is inferred that the combined therapy gives improved outcomes in HIV-BL.  However, this is not what is examined and reported on in Table 2- the 41% is the increase risk of chemo among all stage 3 & 4 BL patients regardless of HIV status.  The risk of death by treatment modality in HIV-BL patients is not reported; nor is it compared to any potential treatment differences seen in non-HIV BL patients.

Similarly for the comparisons of the risk of death among men compared to women; is the risk of death similar or different for male HIV-BL patients to female HIV-BL patients; and how does this compare to what is seen among male non-HIV-BL compared to female non-HIV-BL?

Deprivation is discussed- the authors have data on insurance status but this is not commented on here- whether survival is similar or different by HIV status among those not insured and what if any differences by HIV status there are among the insured.  This would seem most relevant to the study.

Line 236 the authors state they were able to more effectively study survival trends but there are no data presented by time periods.

There is a lack of comparison with relevant literature in the discussion, including the previous NCDB study.

Author Response

Dear Reviewer,

Thank you for your valuable feedback regarding our manuscript, “Long-term survival rates and treatment trends of Burkitt lymphoma in patients with HIV – A National Cancer Database (NCDB) study”

Please see our point-by-point responses below:

Abstract -

1. Stage 3 or 4 BL treated with chemo either alone or with immunotherapy reported in abstract- were these the only cases to be included?

a. Yes, these were the only patients analyzed.

2. In Abstract, the aim is stated as looking at BL mortality.  Please be clear in the abstract whether you are considering overall survival (i.e. death from any cause) or BL-specific survival (i.e. death attributable to BL).

a. Thank you for bringing this to our attention. This has been updated.

3. Please report the HR for the total 5y period so that comparisons can be made to other studies.

a. This is an excellent thought; however, we concluded that presenting an overall HR would have been inappropriate due to the violation of the proportional hazards assumption at the 3-month mark. As such, presenting an overall HR would have been misleading due to the difference in risk of death beginning at the 3-month mark.

4. In abstract, when reporting HR for HIV association with death, please state that this is compared to those without HIV.

a. Thank you, this has been updated.

5. Conclusion of the abstract states there was a difference in risk of death between those treated with chemo & immunotherapy compared to those only on chemo.  Is this for all patients or is there a difference between those with HIV and those without- or indeed are HIV patients doing as well as non-HIV patients on the different treatments?  Please report these results in the abstract.  The introduction suggests that immunotherapy may be beneficial in HIV-BL but this aim is not reflected on in the abstract.

a. Thank you for clarifying this point for us. This has been updated.

Simple summary –

1Please check that the summary is consistent with the aims of the study (eg treatment trends were not examined).  Please change the time period 3-50m to 3- 60m.  Please amend the last sentence in accordance with the comment above – the results are reported on in total and have not examined whether HIV and non-HIV-BL patients have different mortality within treatment modalities.

a. The simple summary has been updated per your comments. Thank you.

Introduction -

1Mention ART in first paragraph-and in the context of US, where treatment accessible

a. Thank you, we agree this is an important point due to the international nature of this publication. It has been updated to included ART recommendations and accessibility in the U.S. per Centers for Disease Control and Prevention (CDC).

2. The introduction would be clearer if it focused on BL. (the info at the start of the second paragraph on NHL and HL is not necessary)

a. Per another reviewer’s recommendation, we have elected to include the context of NHL and HL. However, for better clarity in the introduction, information on BL prevalence and epidemiology has been separated out into its own paragraph and expanded upon.

3. Some more information on treatment of BL and how this has changed- when was rituximab introduced, when was it given to HIV-BL cases?  Is this information only from trials- what is the real-world picture, or is this limited?

a. This is a very clinically relevant point. It is evident in the literature that the inclusion of rituximab into the treatment of BL has improved survival regardless of HIV status. However, the detailed regimens are not included in the NCDB for us to draw conclusions about the effect of rituximab therapy on survival in our study.

Methods -

1. Flow chart- could the flow include the numbers included at each stage of exclusion, so it can be seen how many individuals have stage 3 or 4 BL, how many of those were treated with chemo &/or immunotherapy, etc?

a. Thank you for your comment. However, we believe this information about exclusion was already presented in Figure 1. We have added descriptive counts about stage into Table 1—there was no statistically significant difference in the rates of stage 3 vs 4 between HIV vs no HIV.

2. Cox regression model included comparisons of Black to Whites, Other ethnicities to White but compared Hispanics to not Hispanics.  It would be better to compare Hispanics to Whites also.

a. The NCDB has separate race and ethnicity variables. Unfortunately, there were not enough patients with Hispanic origin to further stratify race and ethnicity (e.g., non-Hispanic white from Hispanic white from non-Hispanic blacks from Hispanic blacks, etc.).

3. Extranodal vs nodal disease were included in the Cox model but these data are not described in Table 1.

a. Thank you for bringing this to our attention. This has been added to Table 1.

Results -

1.Table 1- add a total column to describe the whole cohort of treated stage 3 & 4 BLs.  Add the overall survival and 95% confidence intervals for each group; for Age, this could be given for 2 or 3 sensible age bands; for Sex, include for Males as well as Females. Include numbers for Extra- vs nodal disease.

a. Thank you for your comments. As seen in Table 1, we have addressed this comment. In reference to the overall survival comment, although this would be an excellent point for a case-control study, ours is a cohort study with differential lengths of follow-up resulting in censoring. As such, providing an overall mortality rate across any covariate would not be representative of the true mortality rate given that censoring and constantly-changing denominator.

2. Also add to the first paragraph that those with HIV were less likely to be treated with immunotherapy.

a. This has been updated. Thank you.

3. Paragraph 2 states that the difference in hazard between time periods 0-3m and 3- 60m was statistically significant- not sure that the reporting of HR comparing the periods is helpful and would be better to report the test for proportionality between the HIV and non-HIV survival curves.  Please report the cutpoint on the intervals as either, eg: 0-3m and >3-60m; or 0-<3m and 3-60m as appropriate.  What is the justification for choosing 3 months as the cutpoint; while the Kaplan-Meier curves suggest divergence of the survival curves at 7-8 months?

a. Thank you for your feedback. As stated in our manuscript, the 3-month cutpoint was observed first in the log-negative-log survival curves for HIV status as was subsequently confirmed by model comparison via information criteria (AIC). The separate HRs from 0-3 months and >3-60 months were statistically different, further justifying the cutpoint and need for a heaviside function. In response to your comment about cutpoints observed in the Kaplan-Meier curve, the Cox model assesses risk of death, whereas the Kaplan-Meier method assesses time to death. Further, the Cox model accounted for the correlation inherent to patients treated in the same facility via marginal model (akin to GEE), whereas the more descriptive KaplanMeier survival curves did not. As such, they are different techniques with different outcomes. This has been clarified in the manuscript.

4. Figure 2- please label the x-axis as months since diagnosis.  Please modify the title to include that these are survival curves for stage 3 or 4 Burkitt lymphoma, and add the data source (ie NCDB) and the years of diagnosis (2004-2019).  Please also tidy the figure by removing the title “Product-Limit Survival Estimates”, the title “HIV” from the legend and instead use the labels “non-HIV” instead of 0 and “HIV” instead of 1.

a. This has been updated. Thank you.

5. There is no comparison of the HIV and non-HIV patients within treatment groups which would be a useful addition to examine whether HIV patients on immunotherapy do as well as the non-HIV patients.  If this is something the study could explore, descriptions of the patients within treatment groups by HIV status should be shown too.

a. Thank you for this suggestion. In section 3.1 of the revised manuscript, we have included an HIV status-by-treatment interaction effect. Results indicated that the risk of death between patients with or without HIV differed by treatment from 0-3 months but did not differ after 3 months. Specifically, from 0-3 months in patients who received chemo+immunotherapy, patients who had HIV had 45% lower risk of death. No difference by HIV status was observed from 0-3 months in patients who received only chemo. We have expanded on this finding in our Discussion.

Discussion -

1. The authors mention a previous analysis of HIV-BL mortality using the NCDB for the years 2004-2011.  The present study will cover some of the same patients- more should be said on the overlap, and the HR for patients from 2012-2019 presented.

a. Thank you for this comment. This was updated to better reflect the relationship between our study and the previous NCDB study.

2. The authors later state (line 197) that from 2015 onwards the universal ART and integrase inhibitor-based regimens could be administered with chemo but that their data did not show differences in HIV-BL survival with this treatment modality.  Have the authors examined this?  There were no analyses restricted to 2015 onwards, nor of examining HIV-BL survival by treatment.

a. We have examined this to the extent we could, given that exact ART for patients in the NCDB was not collected. In the third paragraph of the Discussion, we state that, “Beginning in 2015, universal ART and widespread use of potent integrase inhibitor-based regimens were able to be administered concurrently with chemotherapy without major interactions. Surprisingly, our data showed despite the widespread use of this treatment modality, this did not result in significant differences in survival for individuals with HIV-associated Burkitt lymphoma.”

3. Paragraph on lines 205-214 reports that chemo only compared to chemo & immunotherapy increase the risk of death by 41% and then it is inferred that the combined therapy gives improved outcomes in HIV-BL.  However, this is not what is examined and reported on in Table 2- the 41% is the increase risk of chemo among all stage 3 & 4 BL patients regardless of HIV status.  The risk of death by treatment modality in HIV-BL patients is not reported; nor is it compared to any potential treatment differences seen in non-HIV BL patients.

a. Thank you for bringing this to our attention. However, given our newly added results based on your comments above, we believe this issue has been resolved and no longer applies to our study.

4. Similarly for the comparisons of the risk of death among men compared to women; is the risk of death similar or different for male HIV-BL patients to female HIV-BL patients; and how does this compare to what is seen among male non-HIV-BL compared to female non-HIV-BL?

a. Apologies for the confusion, this portion of the discussion has been removed.

5. Deprivation is discussed- the authors have data on insurance status but this is not commented on here- whether survival is similar or different by HIV status among those not insured and what if any differences by HIV status there are among the insured.  This would seem most relevant to the study.

a. We considered including insurance as a co-variate, but the low number of patients within certain insurance groups prevented this and did not want to aggregate into an undifferentiated other category. In the discussion, we included this as a limitation.

6. Line 236 the authors state they were able to more effectively study survival trends but there are no data presented by time periods.

a. Thank you for this feedback. “Trends” was removed from this point in order to properly reflect our study findings.

7. There is a lack of comparison with relevant literature in the discussion, including the previous NCDB study.

a. Thank you for your concern, we hope we have addressed this adequately across our discussion in the revised manuscript.

Reviewer 5 Report

Comments and Suggestions for Authors

The survey on survival of HIV-associated vs. non-HIV-associated Burkitt lymphoma (BL) reports on the tip of the iceberg. The two main results of the study are: HIV-positive status was associated with a significant, 55% increase in risk of death of patients with BL (PWB) between 3-60 months from disease onset, and that chemo-immunotherapy resulted in superior survival when compared with pts treated with chemotherapy alone. As authors list among the limitation of their study, their results are hard to interpret without knowing the cause of death and type and duration of both chemo- and immunotherapy. There are, however, some further weaknesses of the study that need an explanation or, at least mentioning: There are no data what ratio of PWB entered remission and how long was the duration of remission; how long did chemo(-immuno) therapy last (was it or was it not completed within 3 months from diagnosis of BL?); if immunotherapy (presumably anti-CD20 treatment) resulted in B-cell depletion and hypogammaglobulinemia, thus, contributing to immunosuppression caused by HIV infection itself and chemotherapy (?); if pts with HIV-associated BL on (chemo)-immunotherapy received iv or sc immunoglobulin supplementation and if they received IVIG/ScIG in a different cumulative dose than pts with HIV-associated BL on chemotherapy only? Was the intensity of chemotherapy different among pts receiving and not receiving concomitant immunotherapy?

Even despite of the above limitations, mentioned and omitted mentioning by the authors, the manuscript reported – for the first time - on a clinically relevant observation warranting further investigation.

Comments on the Quality of English Language

There are a few typos in the text.

Author Response

Dear Reviewer,

Thank you for your valuable feedback regarding our manuscript, “Long-term survival rates and treatment trends of Burkitt lymphoma in patients with HIV – A National Cancer Database (NCDB) study”

Please see our point-by-point responses below:

The survey on survival of HIV-associated vs. non-HIV-associated Burkitt lymphoma (BL) reports on the tip of the iceberg. The two main results of the study are: HIV-positive status was associated with a significant, 55% increase in risk of death of patients with BL (PWB) between 3-60 months from disease onset, and that chemo-immunotherapy resulted in superior survival when compared with pts treated with chemotherapy alone. As authors list among the limitation of their study, their results are hard to interpret without knowing the cause of death and type and duration of both chemo- and immunotherapy. There are, however, some further weaknesses of the study that need an explanation or, at least
mentioning: There are no data what ratio of PWB entered remission and how long was the duration of remission; how long did chemo(-immuno) therapy last (was it or was it not completed within 3 months from diagnosis of BL?); if immunotherapy (presumably antiCD20 treatment) resulted in B-cell depletion and hypogammaglobulinemia, thus, contributing to immunosuppression caused by HIV infection itself and chemotherapy (?); if pts with HIV-associated BL on (chemo)-immunotherapy received iv or sc immunoglobulin supplementation and if they received IVIG/ScIG in a different cumulative dose than pts with HIV-associated BL on chemotherapy only? Was the intensity of chemotherapy
different among pts receiving and not receiving concomitant immunotherapy?

Even despite of the above limitations, mentioned and omitted mentioning by the authors, the manuscript reported – for the first time - on a clinically relevant observation warranting further investigation.

a. We appreciate your feedback and agree that these are very clinically relevant
comments. Unfortunately, details on cause of death, type and duration of
treatment, remission achievement and duration, timing of BL treatment, and
intensity of BL treatment is not available in the NCDB and not able to be
commented on. We explained this concept as best we could in our limitations,
as well as the need for future studies to answer these questions of great clinical
relevance.

Kind Regards,
Clare Wieland
Primary Author

Round 2

Reviewer 4 Report

Comments and Suggestions for Authors

Thank you for addressing my comments.

There is one outstanding comment which requires attention though and which would add to the manuscript.  The inclusion of survival estimates for the different groups presented in Table 1 would really aid the presentation and interpretation; some of the differences in survival between those with HIV and those without could be explained by differences in the characteristics of HIV compared to non-HIV BL patients (eg age).  To see the potential differences in survival among HIV and non-HIV patients with different characteristics helps to give context to the hazard ratios (I take the point about different censoring times etc but this is true of all studies as well as of the survival curves presented).  It also gives context externally when comparing with other studies.

Author Response

Dear Reviewer, 

Thank you for your feedback on the revisions of our manuscript. 

  1. Thank you for addressing my comments. There is one outstanding comment which requires attention though and which would add to the manuscript.  The inclusion of survival estimates for the different groups presented in Table 1 would really aid the presentation and interpretation; some of the differences in survival between those with HIV and those without could be explained by differences in the characteristics of HIV compared to non-HIV BL patients (eg age).  To see the potential differences in survival among HIV and non-HIV patients with different characteristics helps to give context to the hazard ratios (I take the point about different censoring times etc but this is true of all studies as well as of the survival curves presented).  It also gives context externally when comparing with other studies. 
  1. Thank you for elaborating on this comment. We have included a new table (Table 2) that provides the mortality data as requested.

Reviewer 5 Report

Comments and Suggestions for Authors

I accept your response and the modifications made in the text.

Author Response

Dear Reviewer,

Thank you for your feedback on the revisions of our manuscript. We appreciate the time you have dedicated to our manuscript. 

Round 3

Reviewer 4 Report

Comments and Suggestions for Authors

Thank you for adding the suggested survival estimates.  This really helps understand the patterns of survival in the different groups.

To improve a little further, the title of Table 2 needs to say "Overall survival" instead of "Overall mortality rates" as does the last sentence of the first paragraph of the results.  Please be clear that the numbers in the table relate to the number of deaths and the percents are the overall survival.  Add the confidence intervals for each survival estimate to the table.  It would also be good to start the table with a row describing survival in the total, HIV-ve and HIV+ve BL groups. 

Author Response

Dear Reviewer,

Thank you for your feedback on our revisions.

In addressing your comment,  

  1. To improve a little further, the title of Table 2 needs to say "Overall survival" instead of "Overall mortality rates" as does the last sentence of the first paragraph of the results.  Please be clear that the numbers in the table relate to the number of deaths and the percents are the overall survival.  Add the confidence intervals for each survival estimate to the table.  It would also be good to start the table with a row describing survival in the total, HIV-ve and HIV+ve BL groups. 
  1. Thank you for clarifying your previous comments. We have edited Table 2 to include five-year survival estimates from the Kaplan-Meier method presented alongside 95% log-negative-log confidence intervals. Survival estimates are provided within each demographic and clinical characteristic for overall cohort as well as stratified by HIV status. P-values are based on the log-rank test. 

Kind Regards, 

Clare Wieland 

Primary Author